psychology

maths attainment, school transition, working memory, internalizing symptoms, ALSPAC

**Author for correspondence:**
Danielle Evans
e-mail: de84@sussex.ac.uk

# Internalizing symptoms and working memory as predictors of mathematical attainment trajectories across the primary–secondary education transition

## Danielle Evans, Darya Gaysina and Andy P. Field

School of Psychology, University of Sussex, Brighton, Sussex BN1 9RH, UK

(iD) DE, 0000-0002-5330-3393; APF, 0000-0003-3306-4695

The transition from primary to secondary education is a critical period in early adolescence which is related to increased anxiety and stress, increased prevalence of mental health issues, and decreased maths performance, suggesting it is an important period to investigate maths attainment. Previous research has focused on anxiety and working memory as predictors of maths, without investigating any long-term effects around the education transition. This study examined working memory and internalizing symptoms as predictors of children's maths attainment trajectories (age 7–16) across the transition to secondary education using secondary longitudinal analysis of the Avon Longitudinal Study of Parents and Children (ALSPAC). This study found statistically significant, but very weak evidence for the effect of internalizing symptoms and working memory on maths attainment. Greater parental education was the strongest predictor, suggesting that children of parents with a degree (compared with those with a CSE) gain the equivalent of almost a year's schooling in maths. However, due to methodological limitations, the effects of working memory and internalizing symptoms on attainment cannot be fully understood with the current study. Additional research is needed to further uncover this relationship, using more time-appropriate measures.

# 1. Introduction

The current state of maths attainment and performance of children and adults in the UK is particularly alarming. It is estimated that 49% of working-age adults in the UK have the maths skills expected of primary-school children, with only around 22% of working-age adults, having the equivalent of a C grade or above in GCSE maths [1]. Recent data from the 2015 Programme for International Student Assessment (PISA) suggests that numeracy levels of the UK have not changed since 2012 [2,3], highlighting the increasing importance of investigating factors affecting the maths abilities of children and adolescents, and to intervene before any problems are allowed to extend into adulthood. The problems stemming from poor maths attainment in childhood persist well into adulthood. Underachievement in maths is related to several negative consequences, including lower socioeconomic status (SES) [4] and poor employment prospects through lower rates of full-time employment, more frequent periods of unemployment and lower rates of promotion [5]. Moreover, low maths abilities have been linked to a greater likelihood of living in disadvantaged housing, experiencing homelessness, poorer health outcomes and a higher likelihood of experiencing depression (for a review see [6]). Therefore, the ability to identify predictors of maths attainment as early as possible in childhood provides numerous advantages when attempting to increase maths attainment and improve performance. These benefits apply to both the individual at risk of poor performance, for example, through increased wealth and employment related to increased maths attainment [7], and to the policy-makers implementing strategies that are *effective* in overcoming these issues.

To date, various predictors of maths attainment have been examined, mostly relating to cognitive abilities, though an increasing focus on affective factors is evident in the literature. This study aims to investigate aspects of cognitive and affective domains (and the relationship between them), by focusing on the potential associations between internalizing symptoms, working memory, and maths attainment trajectories from childhood to late adolescence with a particular focus on the period around the transition from primary to secondary education.

## 1.1. Affective factors and maths attainment

Several affective factors such as internalizing symptoms and early temperament have been associated with the development of academic and mathematical abilities, influencing both immediate performance and long-term attainment. Internalizing symptoms refer to self-directed internal distress [8] and include both anxious and depressive experiences. Internalizing symptoms are the most common mental health issue in childhood [9,10], with internalizing disorders (i.e. anxiety and depression) being highly comorbid [11]. Anxiety disorders are usually the first to emerge (age 6 [9]), with around 10% of pre-schoolers aged 2–5 displaying an anxiety disorder [12]. Whereas, other internalizing symptoms (including depression) emerge later in early adolescence [9,13].

Children with increasing trajectories of internalizing symptoms are more likely to experience worse academic achievement compared with their peers with fewer symptoms [14]. The relationship between internalizing symptoms and academic achievement has been found in studies investigating the effects of depression, [15–18], social withdrawal from peers [19] and anxiety [20,21]. Within this literature, the focus of these studies has predominantly been general academic achievement; however, given the importance of maths skills for both children and adults, it is vital we investigate the processes influencing the development and performance of maths abilities specifically. To date, most of the research investigating affective aspects and maths attainment have focused on the role of maths anxiety (see [22] for a review). Nevertheless, the few studies that have examined the impact of general emotional symptoms has found that increased internalizing problems are linked to poor maths skills in pre-school children (aged 3–5 [23]) and adolescents (aged 13–16 [24]), and maths anxiety in adults (aged 18 [25])—which is closely related to maths attainment [22]. However, because Dobbs *et al.* [23] study was cross-sectional, focusing on a limited period in early childhood, and Marcotte *et al.* [24] study was longitudinal with relatively short follow-ups, and focused solely on the effects of depression, it is unknown whether the trajectory of maths attainment is affected long-term by emotional symptoms.

One commonly reported characteristic related to internalizing symptoms is an emotional temperament in early childhood [26,27]. Temperament is broadly defined as an individual's character, personality or traits that remain generally stable over time and across contexts, and is often used as a term

relating to the variation of emotional reactivity displayed early on in children and infants [28]. Those with a particularly 'emotional' temperament experience feelings of fear, anger and sadness, and characteristics closely related to the 'Big Five' [29] personality trait of neuroticism [26]. Therefore, if increased internalizing symptoms are associated with decreased maths attainment, and an emotional temperament predicts internalizing symptoms, then it is plausible that an emotional temperament in childhood could be a very early indicator of poor maths attainment later on in adolescence, which this study aims to explore further. The earlier we can identify potential problems relating to attainment (i.e. by assessing childhood temperament), the earlier interventions can be administered, increasing their effectiveness.

Although there is limited empirical research within this field, we can draw upon the processing efficiency theory (PET) [30] as a framework of how internalizing symptoms (specifically anxiety) might influence the performance of maths skills. The PET proposes that aspects of anxiety (i.e. worry), consume processing and working memory storage resources. The PET assumes that state anxiety is determined by trait anxiety and situational threat, meaning that those experiencing high trait anxiety (and potentially other internalizing symptoms) are more likely to experience high state anxiety in response to highly stressful situations [31]. The intrusive, ruminating thoughts experienced by both anxious and depressed individuals are thought to divert valuable cognitive resources away from the task at hand [32]. From these ideas, we propose that a highly trait-anxious child would experience high state anxiety when taking a maths exam or attempting maths tasks that are perceived as being stressful or 'threatening'. Maths requires mental operations that place a heavy load upon cognitive resources [33]—the same resources that state anxiety is thought to consume, implying that maths is more susceptible to performance interference resulting from high trait and state anxiety compared with other subjects. Additionally, key symptoms of anxious and depressive disorders include the inability to concentrate, loss of interest and worried/suicidal thoughts, all of which, can interfere with academic performance generally. Anxiety also influences a number of school-related factors including school refusal and school dropout [34–36].

## 1.2. Working memory and maths attainment

One key cognitive resource that is often argued to be significant in the relationship between internalizing symptoms and poor academic performance is working memory. Working memory has been referred to as a 'mental workplace in which information can be stored and processed for brief periods of time in the course of demanding cognitive activities' [37, p. 2]. Several researchers have highlighted the importance of working memory within education (see [38]). It has also been viewed as a better predictor of achievement than IQ [39], with several studies linking working memory to academic achievement in children [40,41] and adolescents [37].

Working memory is particularly important for the performance of maths skills, with early measures of working memory (age 4) predicting maths achievement at age 7 [42]. Working memory has been found to account for variance in a number of mathematical skills across different age groups (see [33]), even when controlling for IQ, age and reading skills (e.g. [43]). This is evident in cross-sectional, longitudinal, experimental and maths disabilities studies (for a review see [44]). Additionally, working memory skills can discriminate between students of low and average abilities [37]. Working memory has also been found to correlate with maths anxiety [45], with performance particularly impaired by maths anxiety on high working memory load tasks [46].

There is a growing literature that indicates greater internalizing symptoms predict poorer working memory functioning (for a review see [47]). Internalizing symptoms (anxiety and depression) are related to slower performance on working memory tasks in young children (aged between 3 and 7 [48,49]) and impaired performance in older children (aged between 9 and 11 [50,51]). It is argued that working memory can explain the debilitating effect of anxiety (and potentially depression [32,52]) on maths tasks. Owens *et al.* [53] found that working memory acted as a mediator between anxiety and both maths achievement and general academic achievement [54] in early adolescents (aged 11–13). Further studies suggest that working memory capacity moderates the relationship between anxiety and academic performance: highly anxious individuals with high working memory capacity perform well, whereas highly anxious individuals with low working memory capacity underperform [55]. However, it is unknown whether this relationship persists throughout development, and whether this is specific to anxiety or internalizing symptoms generally, given the similar symptomology and effects on cognitive resources.

## 1.3. The primary–secondary education transition

Research has found that emotional problems are heightened during early adolescence [9,56], are frequently linked to negative outcomes including poor academic achievement [57,58], and increase around the time of the transition from primary to secondary education [59–62]. This transition occurs at age 11 in the UK (other countries vary but mostly transition between ages 10 and 14) and is often regarded as a difficult period for many children [61] that can induce considerable anxiety and stress [59,60]. The transition coincides with a range of developmental changes including puberty and the increased prevalence of mental health disorders [9,56]. In addition, the change to a different educational environment often involves one or more of the following: forming new friendship groups, new school buildings to navigate, establishing *several* new student–teacher relationships and increasing independence from parents. These changes can all be stress-inducing: during the transition, children report greater anxiety over increased workload, relationship concerns, fear of bullies and fear of getting lost [62].

The transition from primary to secondary education is potentially influential in the development of maths abilities for several reasons. Firstly, the transition to secondary education is associated with declines in academic achievement generally and interruptions of attainment growth [63,64]. Secondly, when investigating maths attainment specifically, it is reported that 34% of children do not show any progress in maths during the transition year [65]. Thirdly, several studies show decreasing enjoyment and interest in maths, less involvement in maths class, and decreasing self-efficacy and attitudes towards maths following the transition to secondary education [66–70]. Maths anxiety also increases during the transition, especially for females and high-achievers [71]. Furthermore, decreased maths attainment growth around the transition is associated with increased maths anxiety in adulthood [25]. These findings combined suggest that the primary–secondary education transition is a key period to focus on when investigating predictors of maths attainment.

Overall, the literature presented suggests that the primary–secondary education transition is a stressful period in adolescence that is linked to increased anxiety, poor maths attainment and decreased academic progress. Underachievement in maths is predicted by low working memory capacity and high anxiety, with the PET proposing that state anxiety (which is heightened in individuals with high trait anxiety) consumes working memory resources. Research to date has not investigated the effects of the education transition on maths attainment, specifically examining the role of internalizing symptoms more generally, and working memory capacity on maths attainment trajectories. Empirical studies to date have predominantly focused on the effects of anxiety on maths performance, mostly ignoring other internalizing symptoms that have similar symptoms and effects on attainment and cognition (e.g. depression). Moreover, the existing literature has investigated the effects of working memory and internalizing symptoms (i.e. anxiety) on attainment mostly in relatively small samples of young adolescents, with longitudinal studies often using short-term follow-ups. This dearth of longitudinal research means that we cannot determine how internalizing symptoms and working memory influence the development and performance of maths attainment over time, and if this relationship remains present throughout the entirety of formal schooling, which this study aims to uncover.

## 1.4. The present study

This study aims to investigate the influence of an emotional temperament (measured at age 3), working memory capacity (at age 10) and internalizing symptoms (at age 11) on maths attainment trajectories throughout childhood and adolescence, while investigating other potential contextual predictors including: SES, IQ, biological sex, parental education and traumatic life events. These variables are included as possible contextual predictors based on previous research linking IQ, male sex, higher SES and greater parental education to greater educational attainment generally and within the domain of maths [3,72–74]. This study aims to add to the existing literature by focusing specifically on maths attainment across the transition to secondary education, which has not yet been examined. The present study uses secondary analysis of longitudinal data from the Avon Longitudinal Study of Parents and Children (ALSPAC), examining the trajectory of maths attainment of children from age 7 up to age 16 using national curriculum assessments.

Based on previous research, it is hypothesized that (i) a highly emotional temperament will predict decreased maths attainment; (ii) higher working memory will be related to higher maths attainment; (iii) greater internalizing symptoms will be related to poorer maths attainment; and (iv) working memory will moderate the relationship between internalizing symptoms and maths attainment.

# 2. Method

## 2.1. Sample

The sample consists of participants from the ALSPAC cohort. ALSPAC is a large UK birth cohort of pregnant women residing in the southwest of England with a due date between the 1 April 1991 and the 31 December 1992 [75]. Women were recruited through media campaigns and maternity health services [76]. The core sample consisted of 14 062 live births, of which 13 988 children were alive at 1 year. Additional participants were recruited resulting in a total of 15 589 fetuses, of which 14 901 were alive at 1 year. The sample is generally representative of the overall population; however, there is a slight over-representation of white families with higher SES [76].

Data were collected using self-report postal questionnaires which were completed by the mother, the mother's partners, the study child and the child's teacher. A smaller sub-sample (10%) were invited to attend Children in Focus clinics. The current study uses data from single-child pregnancies, and the first born of twin pregnancies. Children identified as having special educational needs (SEN) at age 7 and/or age 11 and non-English speakers were excluded prior to analysis ($n = 2666$). It was considered that these children may face additional difficulties within education that may influence the results of the study. Due to high rates of attrition and missing data (see Data analysis section), the final sample size was 8769.

The study website contains details of all the data that are available through a fully searchable data dictionary and variable search tool (see http://www.bristol.ac.uk/alspac/researchers/our-data/). All participants provided written informed consent prior to the study. Ethical approval was obtained from the ALSPAC Ethics and Law Committee and the Local Research Ethics Committees. Informed consent for the use of data collected via questionnaires and clinics was obtained from participants following the recommendations of the ALSPAC Ethics and Law Committee at the time.

## 2.2. Measures

### 2.2.1. Maths attainment

In England, children in formal education are assessed at 'key stages' up until age 16. There are four key stages relating to different phases of development with exams or assessments at the end of each stage to evaluate the child's progress. For this study, maths grades at age 6–7 (key stage 1), age 10–11 (key stage 2), age 13–14 (key stage 3) and age 15–16 (key stage 4) were obtained from external education records (National Pupil Database). In key stages 1–3, children are assigned a numerical grade based on their performance ranging from 1 to 8, with a higher grade reflecting higher maths attainment. The expected grades of children for each key stage are level 2 at key stage 1, level 4 at key stage 2 and between levels 5–6 at key stage 3. At key stage 4, adolescents can achieve an alphabetical grade from U to A* which were coded to 2–10, with 10 being the highest grade achievable (i.e. A*). The key outcomes of this study were maths attainment at key stage 2 (age 11), and the overall growth in maths attainment over time.

### 2.2.2. Internalizing symptoms

The strengths and difficulties questionnaire (SDQ) [77] was used to assess children's internalizing symptoms at age 11. The questionnaire assesses general mental health functioning and covers the following areas: prosocial behaviour, hyperactivity, emotional symptoms, conduct problems and peer problems. An internalizing symptoms score was calculated based on a sum of emotional symptoms and peer problems (e.g. 'I am often unhappy' and 'I am usually on my own'). Respondents rated their child's behaviour on a 3-point scale of *not true*, *somewhat true* and *certainly true*. The score ranges from 0 to 20 with a higher score indicating a greater number of difficulties. The SDQ has good concurrent and predictive validity [77], and satisfactory internal consistency (Cronbach's $\alpha$ for emotional difficulties = 0.66, and for peer problems $\alpha = 0.53$) [78].

### 2.2.3. Emotional temperament

The emotionality activity sociability (EAS) temperament measurement scale [79] was included as it is often an early indicator of later emotional and behavioural problems [80], meaning it could be a way

to gauge early internalizing symptoms. The EAS was administered when the child was 38 months old (3.17 years) from which the emotionality scale was extracted to measure an 'emotional temperament'. This score ranges from 5 to 25 with a high score corresponding to a higher level of emotionality. Respondents rated their child's behaviour from *not at all like* to *exactly like* on a 5-point scale. Items included: 'child cries easily', 'child tends to be somewhat emotional', 'child often fusses and cries', 'child gets upset easily' and 'child reacts intensely when upset'.

### 2.2.4. Working memory and general intelligence

IQ was assessed using the Wechsler Intelligence Scale for Children (WISC-III) [81], which was administered as part of a Clinic in Focus session at age 8. Only a sub-sample (10%) of the cohort were invited to this session. The WISC IQ measure comprised five verbal subtests (information, similarities, arithmetic, vocabulary and comprehension) and five performance subtests (picture completion, coding, picture arrangement, block design and object assembly). The short-form of each of the IQ subtests was administered to reduce the length of the session (with the exception of the coding subtest where children completed the full task). The scorings for each of the short-form subtests were then coded to be on the same scale as if the full subtest had been administered. The WISC-III shows good test–retest reliability (0.80–0.89) [82].

Working memory was assessed through a computer task using the Counting Span Task [83] in a Clinic in Focus session at age 10. The Counting Span Task involved presenting children with screens of red and blue dots, followed by asking them to point to and count the number of red dots out loud. Immediately after children counted the dots correctly, they were asked to recall the number of red dots on the screens in the order they were presented. Children were shown two practice screens followed by three sets of two screens, three sets of three screens, three sets of four screens and three sets of five screens, totalling to 42 trials. Children were asked to attempt all the sets regardless of their performance on the set prior. The global score was used representing the number of trials/sets children answered correctly (scored from 0 to 42), reflecting their processing abilities and memory storage.

### 2.2.5. Biological sex

Biological sex at birth was added in as a predictor due to potential sex differences in maths attainment. This variable was dummy coded with females being the reference group. Females accounted for 51.5% of the sample.

### 2.2.6. Socioeconomic status

Socioeconomic status was measured using the Cambridge Social Interaction and Stratification Scale (CAMSIS). The CAMSIS measures occupational structure based upon social interactions [84]. Possible scores can range between 1 (least advantaged) and 99 (most advantaged) with a mean of 50 and a standard deviation of 15 in the national population [85]. The correlation of CAMSIS scores in this sample between mother and partner available at different time points was high (0.6–0.89). In addition, the data available for each time point decreased with each wave, therefore, to retain as much data as possible, the CAMSIS score at 32 weeks gestation was used, with the highest score available taken from either parent.

### 2.2.7. Parental education

Parental education was measured by asking the respondents the highest qualification each parent had achieved at 32 weeks gestation. This was coded into the following six categories: no qualifications, no higher than CSE or GCSE, an O level or equivalent, an A level or equivalent, a teaching or nursing qualification (i.e. vocational qualifications) and a university degree. The highest qualification available was taken from either parent. Of those respondents, 9.2% had the highest qualification of a CSE (this group also included parents with no qualifications due to small numbers), 5.5% had a vocational qualification, 26.3% had an O level, 34.3% had an A level and 24.7% had a degree.

### 2.2.8. Traumatic life events

Traumatic life events were included to control for any events that might increase internalizing symptoms, or decrease maths attainment. Respondents were asked about traumatic events that have happened since the child's fifth birthday, measured when the child was 6.75 years old. This score could range from 0 to 72

and was a weighted sum of 18 items such as 'a pet died', 'the child was separated from their father' and 'the child was sexually abused'. Responses included: *yes the event happened*, and the child was: *very upset*, *quite upset*, *a little upset*, or *not upset*; or *no the event did not happen*. A high score corresponds to a higher number of events, or the child was more negatively affected by the event.

## 2.3. Data analysis

### 2.3.1. Exclusions and missing data

The initial cohort consisted of 13 988 children alive at 1 year. Additional recruitment resulted in 14 901 children alive at 1 year (including singletons and twins; triplets and quadruplets were excluded due to rarity). Withdrawal from the study led to a sample size of 14 684. Of this, data from singletons and the first-born twin were retained for analysis (N = 14 498). Fourteen children were excluded as their first or second main language was not English (N = 14 484). The 2652 children reported to have SEN (identified by teachers at ages 7–8 and 10–11) were excluded leaving a total sample size of 11 832. Finally, 3063 participants lacking data for around 50% of the predictor variables were excluded leaving a final sample size of 8769 (of which 1985 were complete cases).

The longitudinal design of the ALSPAC study resulted in high attrition rates and missing data (see table 1 for missing data patterns and table 2 for a breakdown of missing data per variable). To overcome this, multiple imputation was performed using the semTools [86] and Amelia [87] packages in R [88]. Due to the amount of missing data, 70 imputations were performed and the results were pooled [89]. The outcome variables were included in the imputation model but were not imputed based on best practice [89]. To overcome incomplete outcome data, full information maximum likelihood (FIML) estimation was used. This method uses all available information to produce the maximum likelihood estimation of parameters [90], and has been shown to be a superior method when dealing with missing data [91].

### 2.3.2. Statistical analysis

All analyses were conducted in R v. 3.4.3 [88]. A latent growth model (figure 1) was fit predicting maths attainment trajectories over the transition using the Lavaan package [92]. Maths attainment at 7, 11, 14 and 16 years were endogenous observed variables predicted from latent variables representing the intercept and slope for growth in maths attainment over time. The loadings for the paths from the slope latent variable to the four maths attainment outcomes were constrained to be −4, 0, 3 and 5 so that the intercept represented maths attainment at 11 years old (i.e. the time of the school transition). The predictors specified in the Measures section above were included as exogenous observed variables that predict the intercept and slope of growth in maths attainment. Predictor variables that have no meaningful zero were centred to ease interpretation (SES, IQ, working memory and emotional temperament (EAS)). Emotional temperament, working memory and internalizing symptoms were the substantive predictors of maths attainment in this study. Working memory has been found to moderate the effect of anxiety on cognitive task performance [55]; therefore, the interaction of working memory and internalizing symptoms was included to test for moderation. SES, IQ, gender, parental education and traumatic life events were explored as contextual predictors. When examining parental education, having the highest qualification of a CSE was used as the reference group. The latent growth model provided satisfactory fit indices, CFI = 0.942, TLI = 0.889, RMSEA = 0.108 [90% CI = 0.105, 0.112], SRMR = 0.059. All predictors were entered simultaneously.

# 3. Results

Descriptive statistics for the model predictors and outcomes are presented in table 2. For the outcome of maths attainment, participants' grades were broadly consistent with governmental recommendations for all key stages. Children's yearly progress on average was also in line with governmental guidelines, with children expected to progress half a national curriculum level per year (0.53 levels within this sample).

## 3.1. Predictors of the intercept (maths attainment at age 11)

Table 3 shows the model parameters for predictors of the intercept of maths attainment (i.e. at KS2, age 11). Attainment could vary from level 1 to 8 up until age 14, with most children attaining levels between 3 and 5 at age 11.

**Table 1.** Missing data patterns for predictors of maths attainment. Note: there were 63 other missing data patterns but each one had fewer than 80 cases.

| predictor | 1 | 2 | 3 | 4 | 5 | 6 | 7 | 8 | 9 | 10 | 11 | 12 | 13 | 14 | 15 |
|---|---|---|---|---|---|---|---|---|---|---|---|---|---|---|---|
| working memory | 1 | 0 | 1 | 1 | 1 | 1 | 0 | 0 | 1 | 0 | 0 | 1 | 0 | 0 | 1 |
| SDQ | 1 | 1 | 1 | 0 | 1 | 1 | 1 | 0 | 0 | 0 | 1 | 0 | 0 | 0 | 1 |
| IQ | 1 | 1 | 0 | 1 | 1 | 1 | 0 | 1 | 1 | 0 | 0 | 1 | 0 | 0 | 1 |
| EAS | 1 | 1 | 1 | 1 | 1 | 1 | 1 | 1 | 1 | 1 | 1 | 0 | 1 | 1 | 0 |
| SES | 1 | 1 | 1 | 1 | 1 | 0 | 1 | 1 | 1 | 1 | 0 | 1 | 1 | 0 | 0 |
| parental education | 1 | 1 | 1 | 1 | 1 | 1 | 1 | 1 | 1 | 1 | 1 | 1 | 1 | 1 | 0 |
| biological sex | 1 | 1 | 1 | 1 | 1 | 1 | 1 | 1 | 1 | 1 | 1 | 1 | 1 | 1 | 1 |
| traumatic life events | 1 | 1 | 1 | 1 | 0 | 1 | 1 | 1 | 0 | 1 | 1 | 0 | 0 | 1 | 0 |
| *n* | 3098 | 419 | 267 | 353 | 145 | 562 | 462 | 199 | 95 | 488 | 92 | 85 | 655 | 183 | 168 |

**Table 2.** Summary statistics for the key study measures. WM, working memory; SDQ, internalizing symptoms, EAS, emotionality; MD, missing data.

| measure | n | min | max | *Mdn* | *M* | 95% CI | *s* | MD (%) |
|---|---|---|---|---|---|---|---|---|
| WM | 5669 | 0.00 | 42.00 | 19.00 | 19.14 | [18.95, 19.34] | 57.66 | 34 |
| SDQ | 6010 | 0.00 | 20.00 | 2.00 | 2.40 | [2.34, 2.47] | 6.61 | 30 |
| IQ | 5957 | 49.00 | 151.00 | 105.00 | 106.16 | [105.76, 106.55] | 244.18 | 30 |
| EAS | 7907 | 5.00 | 25.00 | 12.00 | 12.44 | [12.34, 12.53] | 17.40 | 7 |
| SES | 7077 | 23.72 | 99.00 | 58.18 | 58.73 | [58.46, 59.00] | 138.27 | 17 |
| life events | 6925 | 0.00 | 33.00 | 2.00 | 2.94 | [2.86, 3.02] | 11.70 | 19 |
| KS1 maths | 6355 | 0.00 | 3.00 | 2.00 | 2.27 | [2.25, 2.28] | 0.30 | 26 |
| KS2 maths | 6973 | 1.00 | 6.00 | 4.00 | 4.30 | [4.28, 4.31] | 0.51 | 18 |
| KS3 maths | 6037 | 1.00 | 8.00 | 6.00 | 6.18 | [6.15, 6.21] | 1.43 | 29 |
| KS4 maths | 6533 | 2.00 | 10.00 | 7.00 | 7.28 | [7.24, 7.32] | 2.56 | 24 |

**Table 3.** Model parameters for predictors of the intercept of maths attainment. Note: $\beta$ is the standardized parameter estimate.

| predictor | *b* | 95% CI | $\beta$ | *p* |
|---|---|---|---|---|
| working memory | 0.010 | [0.007, 0.014] | 0.106 | 0.000 |
| internalizing (SDQ) | −0.017 | [−0.023, −0.010] | −0.056 | 0.000 |
| working memory × SDQ | 0.001 | [−0.000, 0.002] | 0.028 | 0.066 |
| IQ | 0.019 | [0.018, 0.021] | 0.405 | 0.000 |
| emotionality | 0.000 | [−0.004, 0.004] | −0.001 | 0.956 |
| SES | 0.005 | [0.004, 0.007] | 0.084 | 0.000 |
| education: CSE versus vocational | −0.020 | [−0.101, 0.061] | −0.006 | 0.621 |
| education: CSE versus O level | 0.156 | [0.098, 0.214] | 0.091 | 0.000 |
| education: CSE versus A level | 0.232 | [0.174, 0.289] | 0.146 | 0.000 |
| education: CSE versus degree | 0.412 | [0.344, 0.479] | 0.235 | 0.000 |
| sex | 0.035 | [0.002, 0.068] | 0.023 | 0.040 |
| life events | −0.004 | [−0.009, 0.001] | −0.020 | 0.085 |

Out of the substantive predictors, greater working memory and fewer internalizing symptoms significantly predicted higher maths attainment. For working memory capacity, one additional trial of the task completed correctly equated to a 0.010 increase in maths attainment at age 11 ($p < 0.001$). For internalizing symptoms (SDQ), a 1-unit increase in SDQ score equated to a decrease of 0.017 in attainment ($p < 0.001$). Working memory did not significantly moderate the effect of SDQ on maths attainment ($p = 0.066$). Emotional temperament was not a significant predictor of maths attainment at age 11 ($p = 0.956$).

Of the contextual predictors, higher IQ, higher SES, greater parental education (O level, A level and degree) and male sex were all significant predictors of greater maths attainment. The strongest predictor of attainment was parental education—when compared with parents with a CSE qualification and below, children whose parents had an O level, A level or a degree equated to increased maths attainment at age 11 by 0.156, 0.232 and 0.412 levels, respectively (all *p*-values < 0.001). There was no significant difference in attainment for those whose parents had a vocational qualification compared with a CSE ($b = 0.020$, $p = 0.621$). Males' maths attainment was 0.035 units higher than females' at age 11. A 10-unit increase in IQ equated to a 0.19 increase in maths attainment at age 11. Traumatic life events did not significantly predict maths attainment ($b = 0.004$, $p = 0.085$). Overall, the effects for the significant predictors were all relatively small.

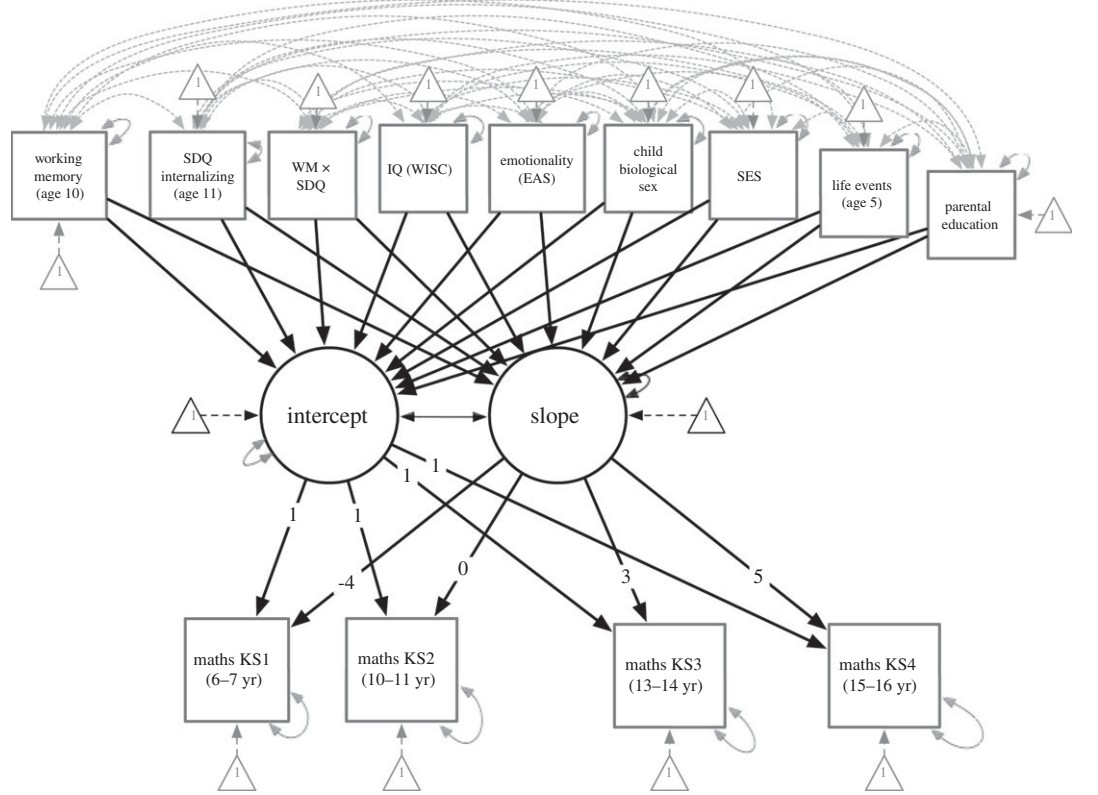

**Figure 1.** Latent growth model for maths attainment trajectories. The intercept represents maths attainment at age 11, and the slope represents maths attainment from age 7 to 16.

## 3.2. Predictors of the slope (maths attainment from age 7 to 16)

Table 4 shows the model parameters for predictors of the slope of maths attainment from age 7 to 16. The group-level growth in maths attainment each year was 0.53, meaning that at average levels of the predictors, children progress by around half a grade level each year.

Out of the substantive predictors, internalizing symptoms ($p = 0.003$) and working memory ($p < 0.001$) were found to significantly (but weakly) predict the rate of change in maths attainment. For internalizing symptoms, a 1-unit increase on the scale decreased the rate of change by 0.002. For an extra trial completed correctly on the working memory task, the rate of change in maths increased by 0.001 units. The moderation of working memory and internalizing symptoms was not found to significantly predict the rate of change in maths attainment ($p = 0.289$). An emotional temperament was not found to predict the rate of change in attainment ($p = 0.892$).

The significant contextual predictors of growth in maths attainment over time were greater IQ, higher SES and greater parental education (O level, A level and degree). Compared with parents with a CSE qualification or below, children whose parents had an O level, A level or a degree experienced increases of 0.023, 0.048 and 0.088, respectively, in the rate of change of their maths attainment per year (all $p$-values $< 0.001$). Children of parents with vocational qualifications did not significantly differ from children of parents with CSE qualifications ($p = 0.817$).

A 10-unit increase in IQ and SES increased the rate of change by 0.03 and 0.01, respectively (both $p$-values $< 0.001$). Neither sex ($p = 0.481$) nor traumatic life events ($p = 0.310$) significantly predicted the rate of change in maths attainment.

It is useful to note that all these effects are relatively small within the context of a group-level rate of change of 0.53 national curriculum levels per year.

## 4. Discussion

This study aimed to identify predictors of maths attainment trajectories across the primary to secondary education transition, specifically investigating: (i) if working memory capacity and internalizing symptoms impact maths attainment pre-transition; (ii) whether working memory capacity and

**Table 4.** Model parameters for predictors of the slope of maths attainment. Note: $\beta$ is the standardized parameter estimate.

| predictor | b | 95% CI | $\beta$ | p |
|---|---|---|---|---|
| working memory | 0.001 | [0.001, 0.002] | 0.094 | 0.000 |
| internalizing (SDQ) | −0.002 | [− 0.003, − 0.001] | −0.042 | 0.003 |
| working memory × SDQ | 0.000 | [− 0.000, 0.000] | 0.020 | 0.289 |
| IQ | 0.003 | [0.002, 0.003] | 0.339 | 0.000 |
| emotionality | 0.000 | [− 0.001, 0.001] | 0.002 | 0.892 |
| SES | 0.001 | [0.001, 0.001] | 0.096 | 0.000 |
| education: CSE versus vocational | 0.002 | [− 0.014, 0.017] | 0.004 | 0.817 |
| education: CSE versus O level | 0.023 | [0.012, 0.034] | 0.082 | 0.000 |
| education: CSE versus A level | 0.048 | [0.037, 0.059] | 0.186 | 0.000 |
| education: CSE versus degree | 0.088 | [0.075, 0.101] | 0.313 | 0.000 |
| sex | 0.002 | [− 0.004, 0.009] | 0.010 | 0.481 |
| life events | −0.001 | [− 0.001, 0.000] | −0.014 | 0.310 |

internalizing symptoms influence the trajectory of maths attainment following the primary–secondary education transition; (iii) whether the effects of internalizing symptoms on maths attainment pre- and post-transition are moderated by working memory capacity, and (iv) if an emotional temperament in childhood is predictive of low maths attainment.

Firstly, the findings partially support the hypothesis with greater working memory ($b = 0.010$, 95% CI [0.007, 0.014]) and fewer internalizing symptoms ($b = –0.017$, 95% CI [–0.023, –0.010]) significantly predicting greater attainment at the time of the transition, suggesting that individuals presenting with greater emotional issues and low working memory capacity in early adolescence are more at risk of poor performance compared with their peers around the transition. However, the effects were small, suggesting that higher IQ, greater parental education and sex (being male) are stronger predictors of higher maths attainment pre-transition. Secondly, the rate of change in maths attainment over time was significantly predicted by working memory (working memory $b = 0.001$, 95% CI [0.001, 0.002]) and internalizing symptoms ($b = –0.002$, 95% CI [–0.003, –0.001]). Increased maths growth was significantly predicted by higher IQ, higher SES and greater parental education, suggesting that children with greater intelligence and higher socioeconomic status progress at a quicker rate across the transition to secondary education compared with their peers. Finally, the interaction between working memory capacity and internalizing symptoms did not significantly predict pre-transition maths attainment ($b = 0.001$), or the trajectory of maths attainment over time ($b = 0.000$), suggesting overall that working memory capacity does not moderate the effects of internalizing symptoms on maths attainment as hypothesized. An additional aim of this study was to explore possible predictors of later internalizing symptoms and whether these would predict attainment. It was hypothesized that an emotional temperament would predict maths attainment, given that childhood temperament and internalizing symptoms are linked [26,27]. However, this idea was not supported (pre-transition $b = 0.000$; trajectory $b = 0.000$), suggesting that we cannot predict later problems with underattainment in maths using emotional difficulties this early on in childhood (age 3).

## 4.1. Internalizing symptoms, working memory and maths attainment

The significant, but extremely small effects of internalizing symptoms on maths attainment pre-transition, and on maths attainment growth in this study is somewhat unexpected. However, previous research has found the reported magnitude of the relationship between internalizing symptoms (anxiety, depression and other emotional issues) and attainment (both general academic and maths) is particularly broad and inconsistent, where some effects range between weak and moderate [14,16–18,20,21,23,24] and others are statistically non-significant ($r$ between − 0.1 and − 0.03 [93]). For example, a meta-analysis by Riglin *et al.* [18] found the relationship between emotional difficulties and school grades to be quite weak; $r = − 0.03$, − 0.10 and − 0.12 for anxiety, internalizing symptoms and depression, respectively. Whereas, Ialongo *et al.* [16] reports odd ratios of achieving a C grade or

worse as 5.71 for males and 3.16 for females. In another study, Riglin *et al.* [17] found a significant association between depressive symptoms and school grades for boys only ($\beta = -0.21$), with no significant relationship found for girls ($\beta = -0.03$). Whereas, Dobbs *et al.* [23] report the effect size to be much more substantial when predicting maths attainment from internalizing symptoms ($\beta = -0.60$). However, these findings further highlight the inconsistencies within the existing literature, due to differences in measurement, ages of participants and the outcome investigated.

There are several practical limitations of this study that require further discussion. Firstly, the measure of internalizing symptoms used in this study consisted of the 'emotional issues' and 'peer problems' subscales of the SDQ (rated by the child's carer), which was assessed when the child was 11 years old. This is important because although anxiety emerges early in childhood (approx. 6 years [9]), depression emerges much later (approx. 13 years [9]), and internalizing symptoms peak around puberty (age 13–15 [13]), potentially suggesting that the time point used to measure internalizing symptoms in this study may have been too early to accurately capture emotional issues. Existing research further highlights that the adolescent period (aged 11–18) is when most psychiatric disorders are diagnosed (approx. 75% [94]), and although an early intervention is vital, it appears that assessing internalizing symptoms this early (age 11) may not be effective for predicting poor maths attainment later on. The variability of internalizing symptoms in this sample was also small ($M = 2.4$, $Mdn = 2.00$, and s.d. = 2.6; possible SDQ scores range from 0 to 20), with slightly lower scores compared with the norms for British children of a similar age [95]. Furthermore, the sample in the present study was more likely to be white with higher SES, whereas in previous research such as Dobbs *et al.* [23] the sample consisted of mostly African-American and Hispanic children, with low SES and parental education (both of which are linked to the greater prevalence of mental disorders [95]). Although internalizing symptoms may correlate moderately with attainment when assessed around the same time in childhood (i.e. [23], $r = 0.29$), the effect on later attainment is either extremely small, or is simply stronger in those with less protective factors such as high SES and parental education.

Previous research has found working memory capacity is linked to better performance on maths tasks from very early childhood through to adulthood [33,42–44,96,97]. However, the findings of the present study only slightly support existing research by finding a very small effect of working memory on maths attainment. One explanation is that the working memory measure used in this study was too blunt to accurately measure the components required for maths tasks. For example, several studies have investigated the predictive power of the separate components of working memory (i.e. central executive, visuo-spatial sketchpad, phonological loop and episodic buffer) on maths attainment and how each system independently accounts for variation in maths performance (e.g. [98]). Researchers have shown that children use different components of working memory when performing maths tasks depending on their age [99–102], with younger children more reliant on visuo-spatial working memory than older children, who tend to use more verbal strategies. The working memory measure in this study consisted of a computer task where children were asked to recall the number of dots presented which taps into visuo-spatial working memory, which explains why this measure only weakly predicts maths attainment at age 11, as the change in strategy from visual to verbal occurs around age 10/11 [103]. However, as children age, they start using more efficient strategies which are less reliant on working memory resources altogether, which also reduces the negative effect of high working memory load on the performance of the maths task [104]. Further to this, children taking maths exams in secondary education in the UK are instructed by their teachers to show their working out to questions to gain 'method marks', meaning that even if the final answer is incorrect, they still receive marks if their technique was otherwise accurate. It could be speculated that this encourages children to write down the information needed to calculate the answer step-by-step, meaning that they do not have to hold this information in their memory for very long, thus reducing the need for high working memory capacity to be successful in these examinations.

This change in memory strategy can also help explain why working memory capacity did not moderate the effects of internalizing symptoms on maths attainment trajectories as found by previous research [53–55]. The PET [30] states that aspects of anxiety negatively affect performance by disrupting the use of cognitive resources (i.e. working memory). Therefore, by switching to strategies that are less susceptible to the negative effects of high working memory load, internalizing symptoms are less likely to interfere with the systems required for successful problem-solving on maths tasks. However, the discrepancies in findings could also be due to differences between studies in sample size, measures of working memory and internalizing symptoms, and the age these variables were measured. These inconsistencies further highlight the complexity of this relationship and the need for additional research to understand the mechanisms involved in maths attainment.

## 4.2. Contextual predictors of maths attainment

There are other notable findings of this study that are worth further discussion. Firstly, as expected, IQ significantly predicted both attainment at the transition ($b = 0.019$), and the rate of growth over time ($b = 0.003$), supported by existing research (e.g. [73]). When interpreting this finding, it should be highlighted that the IQ measure used in this study included tests of arithmetic abilities, which could possibly explain the relatively strong association with maths attainment. However, this idea could not be investigated within this analysis as the individual subscales of the WISC were not available. Secondly, SES was found to predict maths attainment at the transition, and the trajectory over time, though this effect was small (pre-transition $b = 0.005$, post-transition trajectory $b = 0.001$). This finding is not surprising as research has consistently found links between SES and both general academic achievement (for a review see [72]), and maths performance specifically (e.g. [105]). Although, this study provides additional evidence that children with higher SES progress at a very slightly quicker rate compared with their peers.

Thirdly, males were found to achieve significantly higher grades in maths at age 11, but their rate of change over time did not significantly differ from females, suggesting that males start secondary school with a slight grade advantage ($b = 0.035$). Recent data from the PISA assessments support this finding with males on average outperforming females [3], whereas, Lindberg *et al.* [106] found no real gender differences in achievement, and Hyde *et al.* [107] found differences depend on the age of participants and the maths domain assessed. Although, when focusing on the transition, the differences here could be explained by motivational and affective changes occurring around this period of adolescence, such as increasing maths anxiety experienced by females [71] and decreasing maths enjoyment [67]. Given the lack of females in STEM careers [108], further research would be beneficial to improve females' maths performance (and attitudes towards maths) prior to the transition to ensure they start secondary education at the same level as their male peers.

The strongest predictor of maths attainment in this study was greater parental education, significantly predicting higher attainment at the transition and faster growth in maths attainment over time. Compared with children whose parents' highest qualification was a CSE, the highest attainment was for children with parents that had a degree (age 11 $b = 0.412$; trajectory $b = 0.088$), followed by those with A levels (age 11 $b = 0.232$; trajectory $b = 0.048$), then O levels (age 11 $b: = 0.156$; trajectory $b: = 0.023$). There was no significant difference in children's maths attainment between those with parents that had vocational qualifications compared with those with a CSE. The findings here show that children of parents who have a degree (compared with those with a CSE) gain the equivalent of close to an extra year of schooling in attainment at age 11. This finding is not unexpected; however, it demonstrates the importance of parents within their child's education and suggests that having higher-educated parents may potentially 'buffer' the negative impacts of the transition to secondary education on children's attainment.

Parental support is regarded as the most important support system during the transition from primary to secondary education [109,110]. Combined with the findings of this study, this could mean that higher-educated parents support the transition in different ways that lessen the negative impact of the transition on maths attainment. However, it cannot be determined from this study which aspects resulting from greater parental education are the most important for maths attainment during this period. It could be that greater parental education generally is related to living in a 'nice' area, with the child attending a 'good' school for instance. However, given that SES was adjusted for in the model (and ultimately had a very small effect on attainment), it could be speculated that the advantage does not lie within better housing, better schools or monetary resources, but instead may relate to other aspects of parenting such as attitudes towards education, involvement with school activities or helping with homework in a supportive environment. In a review of the education transition, Rens *et al.* [111] highlight the significance of parental support during this period, specifically noting the importance of parental involvement and academic encouragement, and other behaviours such as checking homework, discussing schoolwork and monitoring their child's social wellbeing and academic attainment. It could be that higher-educated parents are more likely to participate in these behaviours, and as such facilitate a more 'successful' transition. Existing research supports this link between greater parental education and more positive school involvement (e.g. [112,113]); however, the effect of these behaviours on maths attainment specifically remain unknown.

Another possible explanation for the effect of parental education on offspring attainment is the passive gene–environment correlation (passive rGE). Passive rGE refers to the phenomenon when exposure to environments depends on an individual's genotype [114,115]. Parents with higher

education provide a more stimulating environment but also pass on genetic factors to their children that can contribute to their higher maths attainment, including traits related to increased motivation, intelligence and temperament, for example [116]. It has been demonstrated that both educational attainment and maths ability have strong genetic components, and may co-occur due to overlapping genetic factors [117,118]. Due to this possible genetic confounding that the current study cannot rule out, additional research is needed to further understand how greater parental education leads to higher maths attainment of children, especially given the potential practical implications this may have for transition intervention strategies.

## 4.3. Study limitations

There are several methodological limitations of this study worth highlighting. One issue with most longitudinal analysis is that the data are now relatively old. Children's emotional, social and educational environments are likely to be very different now compared with the individuals in the ALSPAC sample, meaning that the findings may lack some generalizability.

Another important limitation refers to the measures used within the study. Firstly, the study assumes that working memory and internalizing symptoms remain stable throughout adolescence, as it was not possible to look at changes across the transitioning years and factor this into the analysis. Research has found that visual working memory continues to develop throughout adolescence [119], and the stability of internalizing symptoms is also questionable, with the adolescent period highlighted as a time of considerable disruption [120]. This limitation means that any changes in the development of working memory were not accounted for. Additionally, any changes in emotional difficulties during this period in adolescence may have affected the adaptation to secondary school following the transition [61], which could have affected the participants' maths attainment also. Moreover, recent research has found that changes in internalizing symptoms across this transitionary period predicts increased maths anxiety in adulthood [25], which is associated with maths attainment [22]. This highlights another limitation of the study, which is that a measure of maths anxiety was not available, and would have been beneficial to include within the analysis.

The timing of the measures is also somewhat problematic. In previous studies, working memory and anxiety have been assessed at the same time point—which was not possible within this study. Instead, working memory was measured at age 10, and internalizing symptoms were measured at age 11. This is a limitation because it means that the conclusions drawn from the analysis of the interaction between working memory and internalizing symptoms may be inaccurate due to the delay between assessments. It could be that if working memory and internalizing symptoms were measured simultaneously, the result may have shown a significant moderative effect. However, this is purely speculative and provides a suggestion for future research in this area.

There are additional generalizability issues with the ALSPAC sample that may have affected the strength of the predictors. Specifically, children enrolled in ALSPAC are more likely to achieve higher grades in national curriculum exams (at age 16), and are more likely to be white and of higher socioeconomic background compared with children not in the sample [76]. Therefore, any additional research investigating the effects of the transition should aim to investigate the effects for children with varied backgrounds to ensure validity, especially as children with higher SES tend to adapt to secondary education more successfully (see [61]).

## 4.4. Implications for transition strategies and future research directions

Several interventions have focused on improving academic attainment by increasing working memory capacity; however, very few of these are effective long-term in increasing attainment (e.g. [121,122]). It appears that focusing on other aspects related to attainment might be more suitable. Although not the main aim of this study, the findings imply that strategies focusing on improving parental education (or related factors) for example might be more successful in increasing attainment. Although one possible solution could be to provide better provisions for parents to attain higher educational qualifications, this is not necessarily practical (or desirable) for every family. Furthermore, it has not been determined by this study alone *how* greater parental education leads to greater attainment—first, we must understand the underlying mechanisms linking parental education to better maths attainment before intervention strategies can be proposed and evaluated.

Existing research investigating factors linked to parental education has found that greater parental education is associated with greater school involvement [123], greater enrolment in educational

activities (e.g. music lessons) and higher educational aspirations for their child [116]. To date, few researchers have reviewed the effects of parental interventions focusing on these factors on attainment, and have reported mixed results. Gorard & See [124] found that interventions aiming to increase parental involvement were somewhat promising during pre-school/early primary education, and were ineffective at later stages of schooling. Whereas, another review found that interventions are more beneficial for older children [125]. A report by Ofsted [126] investigating family learning found that interventions provided by adult and community learning hubs in the UK (in schools, Sure Start children's centres and libraries) were successful in helping parents gain further educational qualifications, promoting greater parental involvement in school, and gaining paid employment. However, these programmes are not available beyond key stage 2 (i.e. post-secondary-education transition), meaning it is difficult to say how these programmes would have affected maths performance if they were offered following the transition to secondary education. Furthermore, since Ofsted's review, it is estimated that over 1000 children's centres have been closed [127], with no official figures on the closures of libraries, meaning there are fewer opportunities to encourage parents to participate in these activities, potentially resulting in a wider achievement gap between the children of parents with high and low education.

Although, as discussed previously, it cannot be determined from this study *how* greater parental education is linked to greater offspring attainment, especially so given the possible genetic confounding as highlighted above. Therefore, future research should focus first on what aspects relating to parental education are the most important for maths attainment during the transitional period, taking genetic components into account.

## 4.5. Conclusion

To summarize, this study shows that although internalizing symptoms and low working memory capacity influence maths attainment at the secondary education transition and the growth in attainment over time, these effects are extremely weak at best. Instead, the findings here provide further evidence that low SES and low parental education are two of the biggest risk factors for low attainment in maths prior to the school transition, and continue to affect the rate children progress in secondary education. Additional research is needed to further understand the effect of parental education on maths attainment in adolescents before and after the transition to secondary education to better inform policies and interventions. Recent campaigns launched by the BBC have focused on promoting adult education and maths training in collaboration with the National Numeracy Charity, and while this is a step in the right direction, much more work is needed to overcome the extent of poor numeracy in the UK and the negative effects associated with underachievement in maths.

Ethics. Ethical approval for this research was granted by the University of Sussex Cross-Schools Research Ethics Committee under submission code ER/DE84/1. Ethical approval was obtained from the ALSPAC Ethics and Law Committee and the Local Research Ethics Committees. Informed consent for the use of data collected via questionnaires and clinics was obtained from participants following the recommendations of the ALSPAC Ethics and Law Committee at the time.

Data accessibility. Data used for this submission will be made available on request to the Executive (alspac-exec@bristol.ac.uk). The ALSPAC data management plan (http://www.bristol.ac.uk/alspac/researchers/data-access/documents/alspac-data-management-plan.pdf) describes in detail the policy regarding data sharing, which is through a system of managed open access. Code for pre-processing and analysis is available at: https://osf.io/58kqm/?view_only=f340ae79f311477bb0a973ee201033bf.

Authors' contributions. D.E. and A.P.F. contributed to the design of the study. D.E. analysed and interpreted the data, and drafted the initial manuscript. A.P.F. reviewed the manuscript and data analysis process at all stages. D.G. reviewed the final manuscript and provided feedback and revisions. All authors have given final approval for publication.

Competing interests. We declare we have no competing interests.

Funding. The UK Medical Research Council and the Wellcome Trust (grant ref. 102215/2/13/2) and the University of Bristol provide core support for ALSPAC. A comprehensive list of grants funding is available on the ALSPAC website (http://www.bristol.ac.uk/alspac/external/documents/grant-acknowledgements.pdf). This publication is the work of the authors and they will serve as guarantors for the contents of this paper. This specific research project did not receive any funding.

Acknowledgements. The authors are extremely grateful to all the families who took part in this study, the midwives for their help in recruiting them, and the whole ALSPAC team, which includes interviewers, computer and laboratory technicians, clerical workers, research scientists, volunteers, managers, receptionists and nurses.

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
