## [Reviewer comments · Royal Society Open Science]

Review History

RSOS-191433.R0 (Original submission)

Review form: Reviewer 1

Is the manuscript scientifically sound in its present form?

Yes

Are the interpretations and conclusions justified by the results?

No

Is the language acceptable?

Yes

Do you have any ethical concerns with this paper?

No

Have you any concerns about statistical analyses in this paper?

No

Recommendation?

Major revision is needed (please make suggestions in comments)

Comments to the Author(s)

This study examines predictors of mathematical attainment trajectories, with a focus on the transition from primary to secondary school. Predictors include cognitive, emotional, and demographics. While the paper is very interesting, and contributes to our understanding of maths attainment, after reading the introduction, I found the methodology surprising. Although the methodology are overall good, my concern is that modifications are needed for the introduction to better support/lead to the method and results.

In the method, only maths, not cognition or internalizing behaviours, were examined at the transition from primary to secondary school. The only measurement of emotions in the method are not at primary to secondary school transition, but before schooling (temperament at 3 years) and early schooling ("internalising symptoms" at 6.75 years of age). And working memory is assessed at 8 years of age. However, the key arguments laid out for interest in the transition from primary to secondary school period include anxiety/stress, developmental changes and different educational environment causes greater anxiety etc. Moreover, maths over this transition was of interest because of decreases in enjoyment and self-efficacy and increases in maths anxiety. From my reading, the key arguments for understanding students and their maths performance is due to changes in emotions, education and development. Indeed, throughout the introduction, emotions are discussed in terms of change and trajectory, yet there is no developmental assessment of emotions.

In addition, the authors refer to the Attentional Control Theory to propose why emotions may interact with cognition to affect maths performance. The ACT refers to state anxiety, and is based on the claims that anxiety can affect cognition by affecting attentional control and extends from the Processing Efficiency Theory which suggests worry, the cognitive component of anxiety, takes up WM processing capacity. Can the authors provide a clearer rationale for how they expect internalising symptoms to show long term consequences on cognition and maths performance?

Related, the discussion claims the study found "only weak evidence for the attentional control theory", but this ignores the state nature of the ACT and the time differences in internalising symptoms, cognition and maths assessments. Perhaps this claim is better tested through assessing the relationship between these three factors at a more similar timepoint before adopting a longitudinal design.

No rationale provided in the introduction for examining temperament in early childhood.

My key concern is that the emotional and cognitive assessments were given years apart, and well before the primary to secondary transition. SES, parental education, and IQ are thought to be relatively fixed, and were all found to be significant predictors of age 11 maths and change (i.e., slope and intercept). Conversely, WM and emotions are considered to be less stable over development. In the measurement and analytical approach adopted in the current study, it is possible that these differences in stability and variation in the predictor variables affects the model outcomes. Modelling the intercept at the school transition age, rather than the first assessment of maths may contribute to this. The studies which find the working memory and anxiety affect maths performance/learning tend to assess predictors at a similar time to the maths task (or in longitudinal studies, the model intercept).

It is not clear to me what the IQ measure is. Although the authors specified that the forwards and backwards digitspan were used for WM, no specification is given for the IQ measure. Given there are multiple subtests in the WISC, which subscales were used for IQ and was the focus verbal IQ, performance IQ or both?

Decision letter (RSOS-191433.R0)

02-Jan-2020

Dear Ms Evans,

The editors assigned to your paper ("Internalising Symptoms and Working Memory as Predictors of Mathematical Attainment Trajectories across the Primary-Secondary Education Transition") have now received comments from reviewers. We would like you to revise your paper in accordance with the referee and Associate Editor suggestions which can be found below (not including confidential reports to the Editor). Please note this decision does not guarantee eventual acceptance.

Please submit a copy of your revised paper before 25-Jan-2020. Please note that the revision deadline will expire at 00.00am on this date. If we do not hear from you within this time then it will be assumed that the paper has been withdrawn. In exceptional circumstances, extensions may be possible if agreed with the Editorial Office in advance. We do not allow multiple rounds of revision so we urge you to make every effort to fully address all of the comments at this stage. If deemed necessary by the Editors, your manuscript will be sent back to one or more of the original reviewers for assessment. If the original reviewers are not available, we may invite new reviewers.

- Data accessibility

If you wish to submit your supporting data or code to Dryad (<http://datadryad.org/>), or modify your current submission to dryad, please use the following link:
<http://datadryad.org/submit?journalID=RSOS&manu=RSOS-191433>

- Competing interests

- Authors' contributions

- Acknowledgements

- Funding statement

Kind regards,

Andrew Dunn

on behalf of Dr Emma Hayiou-Thomas (Associate Editor) and Essi Viding (Subject Editor)
openscience@royalsociety.org

Associate Editor's comments (Dr Emma Hayiou-Thomas):

Associate Editor: 1

Comments to the Author:

Thank you for this interesting paper - I enjoyed reading it, and think it has the potential to make a useful contribution to the growing field of mathematical development. The large sample size and sophisticated analytical approach are major strengths, and the writing is clear and readable. However, based on the reviewer's comments, and my own detailed reading, the paper will require some major revisions. The most serious concern the reviewer raised, and with which I strongly agree, is that the theoretical framing in the introduction does not clearly motivate the analysis that is presented. The mechanism by which anxiety and working memory are considered

to affect mathematical attainment is contemporaneous - as the reviewer notes, the ACT refers to state anxiety. The approach you present in the paper, though this is not explicitly stated, assumes stability of both anxiety and WM throughout the primary school years, that will impact maths attainment in the long-term. If that is indeed what you propose, this will need to be much more clearly spelled out and justified in the introduction. Alternatively, modelling the intercept using the Key Stage 1 measure would provide a way to assess these relationships at more similar ages, even if they are not quite concurrent. I would encourage you to consider this alternative model, and/or compare two models with intercepts at Key Stage 1 vs Key Stage 2.

Some more minor points I would also like to see addressed are as follows:

- I was pleased to see a discussion of rGE in the discussion, as this is indeed a major potential confound. The discussion is suitably cautious, but the Abstract doesn't convey this, and will need to be reworded.
- The results section presented the relative effect sizes for the different predictors in the model very clearly. What was harder for the reader to extract was how these compare to the effect sizes reported in the wider literature. It would be helpful to provide this more specific context in the discussion.
- The study limitations should refer to the issue of the relative timing of the measures, and how this relates to the theoretical model; and should also include the lack of a maths anxiety measure. In my view, these are likely to be more serious limitations than the broad generational change in the use of social media.

Comments to Author:

Reviewers' Comments to Author:

Reviewer: 1

Comments to the Author(s)

This study examines predictors of mathematical attainment trajectories, with a focus on the transition from primary to secondary school. Predictors include cognitive, emotional, and demographics. While the paper is very interesting, and contributes to our understanding of maths attainment, after reading the introduction, I found the methodology surprising. Although the methodology are overall good, my concern is that modifications are needed for the introduction to better support/lead to the method and results.

In the method, only maths, not cognition or internalizing behaviours, were examined at the transition from primary to secondary school. The only measurement of emotions in the method are not at primary to secondary school transition, but before schooling (temperament at 3 years) and early schooling ("internalising symptoms" at 6.75 years of age). And working memory is assessed at 8 years of age. However, the key arguments laid out for interest in the transition from primary to secondary school period include anxiety/stress, developmental changes and different educational environment causes greater anxiety etc. Moreover, maths over this transition was of interest because of decreases in enjoyment and self-efficacy and increases in maths anxiety. From my reading, the key arguments for understanding students and their maths performance is due to changes in emotions, education and development. Indeed, throughout the introduction, emotions are discussed in terms of change and trajectory, yet there is no developmental assessment of emotions.

In addition, the authors refer to the Attentional Control Theory to propose why emotions may interact with cognition to affect maths performance. The ACT refers to state anxiety, and is based on the claims that anxiety can affect cognition by affecting attentional control and extends from the Processing Efficiency Theory which suggests worry, the cognitive component of anxiety, takes up WM processing capacity. Can the authors provide a clearer rationale for how they expect internalising symptoms to show long term consequences on cognition and maths performance?

Related, the discussion claims the study found "only weak evidence for the attentional control theory", but this ignores the state nature of the ACT and the time differences in internalising symptoms, cognition and maths assessments. Perhaps this claim is better tested through assessing the relationship between these three factors at a more similar timepoint before adopting a longitudinal design.

No rationale provided in the introduction for examining temperament in early childhood.

My key concern is that the emotional and cognitive assessments were given years apart, and well before the primary to secondary transition. SES, parental education, and IQ are thought to be relatively fixed, and were all found to be significant predictors of age 11 maths and change (i.e., slope and intercept). Conversely, WM and emotions are considered to be less stable over development. In the measurement and analytical approach adopted in the current study, it is possible that these differences in stability and variation in the predictor variables affects the model outcomes. Modelling the intercept at the school transition age, rather than the first assessment of maths may contribute to this. The studies which find the working memory and anxiety affect maths performance/learning tend to assess predictors at a similar time to the maths task (or in longitudinal studies, the model intercept).

It is not clear to me what the IQ measure is. Although the authors specified that the forwards and backwards digitspan were used for WM, no specification is given for the IQ measure. Given there are multiple subtests in the WISC, which subscales were used for IQ and was the focus verbal IQ, performance IQ or both?

Author's Response to Decision Letter for (RSOS-191433.R0)

See Appendix A.

RSOS-191433.R1 (Revision)

Review form: Reviewer 1

Is the manuscript scientifically sound in its present form?

No

Are the interpretations and conclusions justified by the results?

No

Is the language acceptable?

Yes

Do you have any ethical concerns with this paper?

No

Have you any concerns about statistical analyses in this paper?

No

Recommendation?

Major revision is needed (please make suggestions in comments)

Comments to the Author(s)

The authors have made changes to the introduction as well as changed the time of measurement of WM in the model. The analytic model seems quite sound, however, references to mathematics anxiety and the ACT remain problematic.

It seems to me that the two paragraphs starting “Several affective factors” “ to “it is vital we investigate the processes influencing the development and performance of maths abilities specifically” seem to really get at what this research is seeking to do.

The research is useful and important, but the ties between the current methodology and the ACT and mathematics anxiety seem inappropriate. Temperament and internalizing symptoms are general, and can be considered examining how general effective attributes affect education. This is really important. It is very different to a how inhibition and set shifting are affected when someone is experiencing an anxious state. Similarly, mathematics anxiety is different: it is a specific education anxiety. It is thought to be related to mathematics because the anxiety is about mathematics. We would not expect, for example, social anxiety to affect mathematics achievement in the same way.

Drawing in the mathematics anxiety literature and the ACT is problematic: they are discussing very different issues.

Moreover, the description of the ACT is incorrect. It theorizes about inhibition and set shifting, not working memory, and it makes no predictions about the effects of anxiety on mathematics performance. However, the ms states “ Therefore, it would not be unreasonable to suspect that heightened internalising symptoms would generally affect maths attainment in a similar process that the ACT proposes regarding state anxiety and maths performance”

The illustration of the effects of intrusive thoughts (p 5) fits more with the processing efficiency theory

Statements such as “The significant, but extremely small effects of internalising symptoms on maths attainment pre-transition, and on maths attainment growth in this study is somewhat unexpected based upon theory (i.e., the ACT)”. This statement is problematic, given the methodology does not test the theory

Opening paragraph provides little set-up for the paper. This is very general overview into funding and mathematics achievement in the UK. However, the paper focuses on emotional and cognitive predictors of mathematics achievement

Working memory task: This task is not described clearly. Far more information is needed to provide readers with a clear sense of what was displayed, what participants were asked to do, and the measures of task performance. For example, when were participants asked to recall the number of red dots? Was there a delay between viewing the dots and recalling the number of red dots? What was the range of the number of dots? However, from the description that is present, the task seems more of a visual perception task or a short-term visual memory task rather than working memory task. If this is a previously developed task, please cite appropriately. What is the measure of performance? Accuracy. In the results, there is a phrase “One additional trial of the task”: does this mean the task stopped after children got several items wrong in a row?

Also, if you have multiple measures of WM over time, why aren't you utilizing these?

IQ: The IQ assessment included a measure of arithmetic performance. Given early maths achievement/abilities predict later maths achievement, is this IQ-maths relationship just as strong if the arithmetic subscale is taken out of the IQ measure?

Decision letter (RSOS-191433.R1)

10-Mar-2020

Dear Ms Evans:

Manuscript ID RSOS-191433.R1 entitled "Internalising Symptoms and Working Memory as Predictors of Mathematical Attainment Trajectories across the Primary-Secondary Education Transition" which you submitted to Royal Society Open Science, has been reviewed. The comments of the reviewer(s) are included at the bottom of this letter.

Please submit a copy of your revised paper before 02-Apr-2020. Please note that the revision deadline will expire at 00.00am on this date. If we do not hear from you within this time then it will be assumed that the paper has been withdrawn. In exceptional circumstances, extensions may be possible if agreed with the Editorial Office in advance.

In general, we do not allow multiple rounds of revision, and it is therefore unusual for you to be offered a further round of revision. With this in mind, we urge you to make every effort to fully address all of the comments at this stage. If deemed necessary by the Editors, your manuscript will be sent back to one or more of the original reviewers for assessment. If the original reviewers are not available we may invite new reviewers.

- Ethics statement

- Data accessibility

- Competing interests

- Authors' contributions

- Acknowledgements

- Funding statement

on behalf of Dr Emma Hayiou-Thomas (Associate Editor) and Essi Viding (Subject Editor)
openscience@royalsociety.org

Associate Editor Comments to Author (Dr Emma Hayiou-Thomas):

Thank you for your responsiveness to the earlier round of review comments. I think the paper is stronger, particularly now that it includes internalising and WM measures at ages closer to the primary-secondary transition. However, the theoretical framing needs further refinement, as elaborated in Reviewer 1's comments. In particular, the emphasis on the ACT as an explanatory model is problematic in this study given the lack of directly relevant measures; processing efficiency theory may be better aligned with your study aims and measures (I found the work by Owens and colleagues that you cited very helpful in this respect). In your revision, please address each of the remaining concerns, and also consider a supplementary analysis to check whether the strong association with IQ that you find is partly driven by the inclusion of the arithmetic subtest in the WISC. Finally, although your written explanation of the model is clear, a figure would make it even easier for the reader.

Reviewer comments to Author:

Reviewer: 1

Comments to the Author(s)

The authors have made changes to the introduction as well as changed the time of measurement of WM in the model. The analytic model seems quite sound, however, references to mathematics anxiety and the ACT remain problematic.

It seems to me that the two paragraphs starting “Several affective factors” “ to “it is vital we investigate the processes influencing the development and performance of maths abilities specifically” seem to really get at what this research is seeking to do.

The research is useful and important, but the ties between the current methodology and the ACT and mathematics anxiety seem inappropriate. Temperament and internalizing symptoms are general, and can be considered examining how general effective attributes affect education. This is really important. It is very different to a how inhibition and set shifting are affected when someone is experiencing an anxious state. Similarly, mathematics anxiety is different: it is a specific education anxiety. It is thought to be related to mathematics because the anxiety is about mathematics. We would not expect, for example, social anxiety to affect mathematics achievement in the same way.

Drawing in the mathematics anxiety literature and the ACT is problematic: they are discussing very different issues.

Moreover, the description of the ACT is incorrect. It theorizes about inhibition and set shifting, not working memory, and it makes no predictions about the effects of anxiety on mathematics performance. However, the ms states “ Therefore, it would not be unreasonable to suspect that heightened internalising symptoms would generally affect maths attainment in a similar process that the ACT proposes regarding state anxiety and maths performance”

The illustration of the effects of intrusive thoughts (p 5) fits more with the processing efficiency theory

Statements such as “The significant, but extremely small effects of internalising symptoms on maths attainment pre-transition, and on maths attainment growth in this study is somewhat unexpected based upon theory (i.e., the ACT)”. This statement is problematic, given the methodology does not test the theory

Opening paragraph provides little set-up for the paper. This is very general overview into funding and mathematics achievement in the UK. However, the paper focuses on emotional and cognitive predictors of mathematics achievement

Working memory task: This task is not described clearly. Far more information is needed to provide readers with a clear sense of what was displayed, what participants were asked to do, and the measures of task performance. For example, when were participants asked to recall the number of red dots? Was there a delay between viewing the dots and recalling the number of red dots? What was the range of the number of dots? However, from the description that is present, the task seems more of a visual perception task or a short-term visual memory task rather than working memory task. If this is a previously developed task, please cite appropriately. What is the measure of performance? Accuracy. In the results, there is a phrase “One additional trial of the task”: does this mean the task stopped after children got several items wrong in a row?

Also, if you have multiple measures of WM over time, why aren't you utilizing these?

IQ: The IQ assessment included a measure of arithmetic performance. Given early maths achievement/abilities predict later maths achievement, is this IQ-maths relationship just as strong if the arithmetic subscale is taken out of the IQ measure?

Author's Response to Decision Letter for (RSOS-191433.R1)

See Appendix B.

Decision letter (RSOS-191433.R2)

03-Apr-2020

Dear Ms Evans,

It is a pleasure to accept your manuscript entitled "Internalising Symptoms and Working Memory as Predictors of Mathematical Attainment Trajectories across the Primary-Secondary Education Transition" in its current form for publication in Royal Society Open Science. The comments of the Editor who reviewed your manuscript are included at the foot of this letter.

on behalf of Dr Emma Hayiou-Thomas (Associate Editor)
openscience@royalsociety.org

Associate Editor Comments to Author (Dr Emma Hayiou-Thomas):

Thank you for your responsiveness to the review comments. I think the theoretical framing of the paper works much better in its current form, and sets the study up nicely.

Appendix A

Author response to reviews of

Manuscript RSOS-191433

Internalising Symptoms and Working Memory as Predictors of Mathematical Attainment Trajectories across the Primary-Secondary Education Transition

submitted to *Royal Society Open Science*

RC: Reviewer Comment AR: Author Response Manuscript text

Dear Dr Hayiou-Thomas,

Thank you very much for taking the time to consider our manuscript for publication at *Royal Society Open Science*. In the following we address your own and the reviewer's concerns point-by-point, and describe the revisions made.

1. Reviewer #1 (Dr Hayiou-Thomas)

RC: The most serious concern the reviewer raised, and with which I strongly agree, is that the theoretical framing in the introduction does not clearly motivate the analysis that is presented. The mechanism by which anxiety and working memory are considered to affect mathematical attainment is contemporaneous - as the reviewer notes, the ACT refers to state anxiety. The approach you present in the paper, though this is not explicitly stated, assumes stability of both anxiety and WM throughout the primary school years, that will impact maths attainment in the long-term. If that is indeed what you propose, this will need to be much more clearly spelled out and justified in the introduction. Alternatively, modelling the intercept using the Key Stage 1 measure would provide a way to assess these relationships at more similar ages, even if they are not quite concurrent. I would encourage you to consider this alternative model, and/or compare two models with intercepts at Key Stage 1 vs Key Stage 2.

AR: Thanks for this suggestion, it was a really good point and luckily we had later measures available for working memory and internalising symptoms in ALSPAC that allowed us to address this directly. The model now focuses on working memory at age 10 and internalising symptoms at age 11 to be closer to the primary-secondary education transition. Generally, this change did not alter the findings substantially, the betas for working memory and internalising symptoms are very slightly smaller than before, and the interaction between working memory and internalising symptoms is no longer statistically significant (though ultimately, the betas are essentially the same as the model before for this interaction).

The introduction now includes a rationale on the examination of the long-term effects of internalising symptoms, by highlighting the relationship between trait and state anxiety and the shared symptomology/effects on cognition. A discussion of the stability of the predictors and timing of them is included in the limitations within the discussion section.

RC: I was pleased to see a discussion of rGE in the discussion, as this is indeed a major potential confound. The discussion is suitably cautious, but the Abstract doesn't convey this, and will need to be reworded.

AR: Abstract has been reworded to:

However, due to methodological limitations, the effects of working memory and internalising symptoms on attainment cannot be fully understood with the current study. Additional research is needed to further uncover this relationship, utilising more time-appropriate measures.

RC: The results section presented the relative effect sizes for the different predictors in the model very clearly. What was harder for the reader to extract was how these compare to the effect sizes reported in the wider literature. It would be helpful to provide this more specific context in the discussion.

AR: A discussion of the effect sizes of internalising symptoms on attainment is now included in the discussion (p. 20-21).

RC: The study limitations should refer to the issue of the relative timing of the measures, and how this relates to the theoretical model; and should also include the lack of a maths anxiety measure. In my view, these are likely to be more serious limitations than the broad generational change in the use of social media.

AR: These have been included as limitations in the discussion, removing the in-depth discussion of the changes in children's environment and change in social media use (p. 26-27).

2. Reviewer #2

RC: I found the methodology surprising. Although the methodology are overall good, my concern is that modifications are needed for the introduction to better support/lead to the method and results.

In the method, only maths, not cognition or internalizing behaviours, were examined at the transition from primary to secondary school. The only measurement of emotions in the method are not at primary to secondary school transition, but before schooling (temperament at 3 years) and early schooling ("internalising symptoms" at 6.75 years of age). And working memory is assessed at 8 years of age. However, the key arguments laid out for interest in the transition from primary to secondary school period include anxiety/stress, developmental changes and different educational environment causes greater anxiety etc. Moreover, maths over this transition was of interest because of decreases in enjoyment and self-efficacy and increases in maths anxiety. From my reading, the key arguments for understanding students and their maths performance is due to changes in emotions, education and development. Indeed, throughout the introduction, emotions are discussed in terms of change and trajectory, yet there is no developmental assessment of emotions.

AR: As mentioned above, the model has been changed to include measures of working memory and internalising symptoms as close to the transition as is available in the dataset. Greater discussion on the inclusion of temperament has been added to the introduction (p. 6-7). It was not possible to examine the change in trajectory for emotions and cognition, as comparable measures were unfortunately not available in the ALSPAC dataset for this time period, but this has been highlighted as a limitation in the discussion section.

RC: In addition, the authors refer to the Attentional Control Theory to propose why emotions may interact

with cognition to affect maths performance. The ACT refers to state anxiety, and is based on the claims that anxiety can affect cognition by affecting attentional control and extends from the Processing Efficiency Theory which suggests worry, the cognitive component of anxiety, takes up WM processing capacity. Can the authors provide a clearer rationale for how they expect internalising symptoms to show long term consequences on cognition and maths performance? Related, the discussion claims the study found "only weak evidence for the attentional control theory", but this ignores the state nature of the ACT and the time differences in internalising symptoms, cognition and maths assessments. Perhaps this claim is better tested through assessing the relationship between these three factors at a more similar timepoint before adopting a longitudinal design.

AR: A rationale has been added to the introduction relating to the expectation that internalising symptoms will affect maths performance long-term based upon the relationship between trait and state anxiety, and the similarities in symptoms between state anxiety and internalising symptoms (p. 5-6). The state nature of the ACT is highlighted further in the discussion section also.

RC: No rationale provided in the introduction for examining temperament in early childhood.

AR: A rationale is now included in the introduction, see pages 6-7.

RC: My key concern is that the emotional and cognitive assessments were given years apart, and well before the primary to secondary transition. SES, parental education, and IQ are thought to be relatively fixed, and were all found to be significant predictors of age 11 maths and change (i.e., slope and intercept). Conversely, WM and emotions are considered to be less stable over development. In the measurement and analytical approach adopted in the current study, it is possible that these differences in stability and variation in the predictor variables affects the model outcomes. Modelling the intercept at the school transition age, rather than the first assessment of maths may contribute to this. The studies which find the working memory and anxiety affect maths performance/learning tend to assess predictors at a similar time to the maths task (or in longitudinal studies, the model intercept).

AR: Working memory and internalising symptoms are now measured as close to the transition as possible in the ALSPAC dataset. However, this still assumes the stability of these measures which is included as a limitation in the discussion (p. 26-27).

RC: It is not clear to me what the IQ measure is. Although the authors specified that the forwards and backwards digitspan were used for WM, no specification is given for the IQ measure. Given there are multiple subtests in the WISC, which subscales were used for IQ and was the focus verbal IQ, performance IQ or both?

AR: The description of the IQ measure in the method section now states the subtests used:

The WISC IQ measure comprised of five verbal subtests (information, similarities, arithmetic, vocabulary and comprehension) and five performance subtests (picture completion, coding, picture arrangement, block design and object assembly). The short-form of each of the IQ subtests were administered to reduce the length of the session (with the exception of the coding subtest where children completed the full task). The scorings for each of the short-form subtests were then coded to be on the same scale as if the full subtest had been administered.

We hope that you find these revisions responsive to your recommendations, and we look forward to hearing from you in due course.

Yours sincerely

Danielle Evans

Appendix B

Author response to reviews of

Manuscript RSOS-191433.R1

Internalising Symptoms and Working Memory as Predictors of Mathematical Attainment Trajectories across the Primary-Secondary Education Transition

submitted to *Royal Society Open Science*

RC: Reviewer Comment AR: Author Response Manuscript text

Dear Dr Hayiou-Thomas,

Thank you very much for taking the time to reconsider our revised manuscript for publication at *Royal Society Open Science*. In the following we address your own and the reviewer's concerns and suggestions point-by-point, and describe the revisions made to the manuscript.

1. Reviewer #1 (Dr Hayiou-Thomas)

RC: The theoretical framing needs further refinement, as elaborated in Reviewer 1's comments. In particular, the emphasis on the ACT as an explanatory model is problematic in this study given the lack of directly relevant measures; processing efficiency theory may be better aligned with your study aims and measures (I found the work by Owens and colleagues that you cited very helpful in this respect).

AR: The discussion of the ACT has been removed from all sections and the processing efficiency theory (PET) has been added as a theoretical framework to the introduction and discussion (page 6 & 23).

RC: consider a supplementary analysis to check whether the strong association with IQ that you find is partly driven by the inclusion of the arithmetic subtest in the WISC

AR: A supplementary analysis looking at the association between the arithmetic subtests of the IQ measure was not possible as we do not have access to this specific variable in the ALSPAC dataset - we only have the total IQ score. However, this has been noted in the discussion on page 23 as below.

When interpreting this finding, it should be highlighted that the IQ measure used in this study included tests of arithmetic abilities, which could possibly explain the relatively strong association with maths attainment. However, this idea could not be investigated within this analysis as the individual subscales of the WISC were not available.

RC: although your written explanation of the model is clear, a figure would make it even easier for the reader.

AR: A figure of the growth model has been added to the manuscript (page 53).

2. Reviewer #2

- RC:** The research is useful and important, but the ties between the current methodology and the ACT and mathematics anxiety seem inappropriate. Temperament and internalizing symptoms are general, and can be considered examining how general effective attributes affect education. This is really important. It is very different to a how inhibition and set shifting are affected when someone is experiencing an anxious state. Similarly, mathematics anxiety is different: it is a specific education anxiety. It is thought to be related to mathematics because the anxiety is about mathematics. We would not expect, for example, social anxiety to affect mathematics achievement in the same way. Drawing in the mathematics anxiety literature and the ACT is problematic: they are discussing very different issues. Moreover, the description of the ACT is incorrect. It theorizes about inhibition and set shifting, not working memory, and it makes no predictions about the effects of anxiety on mathematics performance. However, the ms states "Therefore, it would not be unreasonable to suspect that heightened internalising symptoms would generally affect maths attainment in a similar process that the ACT proposes regarding state anxiety and maths performance". The illustration of the effects of intrusive thoughts (p 5) fits more with the processing efficiency theory. Statements such as "The significant, but extremely small effects of internalising symptoms on maths attainment pre-transition, and on maths attainment growth in this study is somewhat unexpected based upon theory (i.e., the ACT)". This statement is problematic, given the methodology does not test the theory.
- AR:** As mentioned above, the discussion of the ACT has now been replaced with a discussion of the PET, highlighting the effects of state and trait anxiety on working memory capacity and subsequent performance. The two statements included above have also been removed from the manuscript.
- RC:** Opening paragraph provides little set-up for the paper. This is very general overview into funding and mathematics achievement in the UK. However, the paper focuses on emotional and cognitive predictors of mathematics achievement.
- AR:** The opening paragraph has been revised - the discussion of funding has been removed but still includes some of the outcomes associated with low maths attainment in the UK to highlight the importance of investigating predictors of maths attainment.
- RC:** Working memory task: This task is not described clearly. Far more information is needed to provide readers with a clear sense of what was displayed, what participants were asked to do, and the measures of task performance. For example, when were participants asked to recall the number of red dots? Was there a delay between viewing the dots and recalling the number of red dots? What was the range of the number of dots? However, from the description that is present, the task seems more of a visual perception task or a short-term visual memory task rather than working memory task. If this is a previously developed task, please cite appropriately. What is the measure of performance? Accuracy. In the results, there is a phrase "One additional trial of the task": does this mean the task stopped after children got several items wrong in a row?
- AR:** A more thorough description of the working memory measure has been included on pages 13-14 (see below), hopefully this provides answers to the above (although it should be noted, from the documentation in ALSPAC it is unknown how many dots were presented at a time).

Working memory was assessed through a computer task using the Counting Span Task (Case, Kurland, & Goldberg, 1982) in a Clinic in Focus session at age 10. The Counting Span Task involved presenting children with screens of red and blue dots, followed by asking them to point to and count the number of red dots out loud. Immediately after children counted the dots correctly, they were asked to recall the number of red dots on the screens in the order they were presented. Children were shown 2 practice screens followed by three sets of two screens, three sets of three screens, three sets of four screens, and three sets of five screens, totalling to 42 trials. Children were asked to attempt all the sets regardless of their performance on the set prior. The global score was used representing the number of trials/sets children answered correctly (scored from 0-42), reflecting their processing abilities and memory storage.

RC: Also, if you have multiple measures of WM over time, why aren't you utilizing these?

AR: The other measures of working memory have not been included as there are no corresponding measures of anxiety/internalising symptoms at similar timepoints.

RC: The IQ assessment included a measure of arithmetic performance. Given early maths achievement/abilities predict later maths achievement, is this IQ-maths relationship just as strong if the arithmetic subscale is taken out of the IQ measure?

AR: Unfortunately we do not have access to the individual items of the WISC IQ measure so cannot assess this directly. However, as stated above, this has been added into the discussion as being a possibility that should be considered when interpreting this finding (page 23).

We are extremely appreciative of your second round of reviews and hope that these revisions are responsive to your recommendations.

We look forward to hearing from you in due course.

Yours sincerely

Danielle Evans